# Complexes of Gold(III) with Hydrazones Derived from Pyridoxal: Stability, Structure, and Nature of UV-Vis Spectra

**DOI:** 10.3390/ijms25095046

**Published:** 2024-05-06

**Authors:** Natalia N. Kuranova, Oleg A. Pimenov, Maksim N. Zavalishin, George A. Gamov

**Affiliations:** Department of General Chemical Technology, Ivanovo State University of Chemistry and Technology, Sheremetevskii pr. 7, Ivanovo 153000, Russia; kuranova_nn@isuct.ru (N.N.K.); pimenov@isuct.ru (O.A.P.); zavalishin00@gmail.com (M.N.Z.)

**Keywords:** gold(III), hydrazone, pyridoxal, stability constant, speciation, DFT, molecular orbitals, UV-Vis spectra

## Abstract

Pyridoxal and pyridoxal 5′-phosphate are aldehyde forms of B_6_ vitamin that can easily be transformed into each other in the living organism. The presence of a phosphate group, however, provides the related compounds (e.g., hydrazones) with better solubility in water. In addition, the phosphate group may sometimes act as a binding center for metal ions. In particular, a phosphate group can be a strong ligand for a gold(III) ion, which is of interest for researchers for the anti-tumor and antimicrobial potential of gold(III). This paper aims to answer whether the phosphate group is involved in the complex formation between gold(III) and hydrazones derived from pyridoxal 5′-phosphate. The answer is negative, since the comparison of the stability constants determined for the gold(III) complexes with pyridoxal- and pyridoxal 5′-phosphate-derived hydrazones showed a negligible difference. In addition, quantum chemical calculations confirmed that the preferential coordination of two series of phosphorylated and non-phosphorylated hydrazones to gold(III) ion is similar. The preferential protonation modes for the gold(III) complexes were also determined using experimental and calculated data.

## 1. Introduction

Pyridoxal, pyridoxamine, pyridoxine, and their 5′-phosphates (Figure 1) are mutually interconvertible in the living organism and united under the code name B_6_ vitamin [1]. These compounds are coenzymes in several dozens of the classified biochemical activities and regulate important processes such as decarboxylation, deamination, and transamination [2]. Both aldehyde forms of B_6_ vitamins, pyridoxal (**PL**), and pyridoxal 5′-phosphate (**PLP**), can form various Schiff bases and hydrazones that can manifest their own biological activity. In particular, the hydrazone formed by **PL** and isoniazid, a well-known anti-tubercular agent, (**PIH**) has long been the subject of extensive research as a potent drug whose biological activity follows from iron-chelating properties [3,4,5,6,7]. The anti-proliferative [8] and thalassemia treatment [9] effects were identified as being among the most promising actions of **PIH**. In addition, a ferric complex of **PIH** can provide reticulocytes with iron(III) for heme biosynthesis [10,11]. Anti-oxidative action of **PL**-derived hydrazones was also reported [12,13,14,15,16]. This could be due to the binding of bivalent metal ions into stable complexes, thus preventing their participation in Fenton-like reactions that lead to the generation of reactive oxygen species. However, ferric complexes of **PL**-derived hydrazones may have cytotoxic effects because of the induction of oxidative stress [17,18]. In general, hydrazones are versatile compounds gaining attention not only because of their biological activity, but also for their catalytic and fluorescent properties, which can be used for metal recognition and glass and phosphor production [19,20,21].

**PLP**-derived hydrazones have received less attention with regard to their biological activity than **PL** hydrazones. However, the biochemical properties of the phosphorylated hydrazones should be similar to those of **PL** derivatives. The phosphate group located in the periphery of the molecule does not significantly influence the chelating properties (it is noteworthy that, in general, the peripheral substituents of the hydrazones can be changed significantly without the loss of the most important properties; see, e.g., papers [22,23,24,25,26]). The hydrazone formed by **PLP** and isoniazid can also show anti-oxidant properties, because it inhibits the copper(II)-mediated oxidation of ascorbic acid due to the ability of Cu^2+^ [27] binding. Comparisons of the stability constants of the complexes formed by Ni^2+^, Zn^2+^ ions, and **PL-** and **PLP**-derived hydrazones [28,29] show that introducing the phosphate group leads to negligible changes in the value of the decimal logarithm of the stability constant (log β). In living organisms, hydrazones of pyridoxal 5′-phosphate can form **PL**-derived compounds as a result of enzymatic dephosphorylation [30]. This process may alter the solubility of hydrazones and their metal complexes, as the phosphate group bearing two protons that can dissociate easily is eliminated from the molecule.

Being low-toxic and membranotropic [31,32,33], **PL**-derived hydrazones seem suitable for use in the preparation of metal complexes with biological activity, such as Pt^2+^ or Au^3+^ complexes, as they would help metal ions in permeating the cell membrane. Gold(III) complexes are of special interest, as they are isostructural and isoelectronic to cisplatin, a well-known anti-tumor drug. Hydrazones such as the N,O-donor ligands are a good choice for gold(III), in particular as Au^3+^’s Lewis acidity corresponds well to the nitrogen and oxygen’s Lewis basicity [34]. Different complexes of gold(III) have shown potential anti-tumor [35,36] and antibacterial [37] effects. However, unlike gold(I) compounds that have long been used for arthritis therapy [38], gold(III) compounds are not yet among the approved pharmaceuticals.

Our previous papers [39,40] were devoted to the study of the stability and structure of several gold(III) complexes with hydrazones derived from pyridoxal 5′-phosphate. However, some questions remained unanswered.

First, is the phosphate group involved in the complex formation? Can the partial negative charge accumulated on the phosphate group make it attractive for the cations (including gold(III) species) [40]? This possibility cannot be neglected, as both insoluble [41] and soluble [42] complexes of gold(III) with inorganic phosphate are known, including a valuable histochemical label, so-called black-gold [43,44,45].

Second, the calculated UV-Vis spectrum of the single protonated complexes of gold(III) with hydrazones has shown the worst agreement with the experimental spectra [40]. What is the reason behind this discrepancy?

Therefore, the present contribution aims to 1) clarify the possible role of the phosphate group in the complexation of B_6_ vitamin-derived hydrazones with gold(III) species. To figure out the phosphate effect, it is convenient to study the complexation of gold(III) with pyridoxal hydrazones that are analogous to the **PLP** derivatives described earlier [39]. 2) Achieve a better agreement between the calculated and experimental UV-Vis spectra for different protonated complexes. The reconsideration of the protonation order is required to solve the problem. 3) Obtain new data on the stability of the complexes of gold(III) and hydrazones derived from pyridoxal. The final goal is to obtain several hydrazones that can be used for the synthesis of gold(III) complexes with antimicrobial and anti-tumor potential. Despite small difference in structure (in the presence or absence of phosphate groups), either **PL**- or **PLP**-derived hydrazones may be more beneficial in the end in terms of their biological effect).

The assumed general structural formula of the gold(III) complex with different hydrazones, with an indication of the hypothesized protonation sites, is given in Figure 2.

## 2. Results and Discussion

### 2.1. Stability of Gold(III) Complexes with Hydrazones

The reasoning described in our previous paper [39] devoted to the gold(III) complexation with **PLP**-derived hydrazones is also applicable to the complexation with pyridoxal-derived ligands. During the titration, the pH value changed from 3.3 to 9.8 units, which did not cause the precipitate formation.

Examples of spectrophotometric titration are given in Figure 3a,b and Appendix A. These spectral data (similar to those obtained in the previous paper [39]) were used for calculating the stability constants of the gold(III) complexes formed.

The following processes of the general formula mH^+^ + nL^2−^ + p[AuCl_4_]^−^ + qOH^−^ ⇄ [(AuCl)_p_(OH)_q_(H_m_L_n_)]^m+2n+p+q^ (the overall stability constant for this process is expressed by the following equation: β = [[(AuCl)_p_(OH)_q_(H_m_L_n_)]^m+2n+p+q^] · [H^+^]^−m^ · [L^2-^]^−n^ · [[AuCl_4_]^−^]^−p^ [OH^−^]^−n^) should be set in the stoichiometric scheme for calculations of stability constants using KEV software [46]:H^+^ + L^2^− ⇄ HL^−^(1)
2H^+^ + L^2^− ⇄ H_2_L(2)
3H^+^ + L^2^− ⇄ H_3_L^+^(3)
[AuCl_4_]^−^ + OH^−^ ⇄ [AuCl_3_OH]^−^ + Cl^−^(4)
[AuCl_4_]^−^ + 2OH^−^ ⇄ [AuCl_2_(OH)_2_]^−^ + 2Cl^−^(5)
[AuCl_4_]^−^ + 3OH^−^ ⇄ [AuCl(OH)_3_]^−^ + 3Cl^−^(6)
[AuCl_4_]^−^ + 4OH^−^ ⇄ [Au(OH)_4_]^−^ + 4Cl^−^(7)
H^+^ + OH^−^ ⇄ HOH(8)
[AuCl_4_]^−^ + L^2−^ ⇄ [AuClL] + 3Cl^−^(9)
[AuCl_4_]^−^ + H^+^ + L^2−^ ⇄ [AuCl(HL)]^+^ + 3Cl^−^(10)
[AuCl_4_]^−^ + 2H^+^ + L^2−^ ⇄ [AuCl(H_2_L)]^2+^ + 3Cl^−^(11)

The constants of Processes (1)–(3) are adopted from [47] for **PL-INH** (log β_1_ = 11.01; log β_2_ = 19.32; log β_3_ = 25.38), **PL-F2H** (log β_1_ = 11.42; log β_2_ = 17.94; log β_3_ = 21.81), **PL-F3H** (log β_1_ = 11.97; log β_2_ = 17.45; log β_3_ = 22.25), **PL-T2H** (log β_1_ = 11.83; log β_2_ = 17.57; log β_3_ = 21.57), and **PL-T3H** (log β_1_ = 12.0; log β_2_ = 18.25; log β_3_ = 21.65). The constants of Processes (4)–(7) are taken from the paper [48] (log β_(4)_ = 7.87; log β_(5)_ = 14.79; log β_(6)_ = 20.92; log β_(7)_ = 25.98). The value of log β_(8)_ = 13.91 was taken from the report [49]. The values of log β (Equations (9)–(11)) are to be determined.

As in the previous paper [39], the different stoichiometric models were tested to find the most suitable one. They included the formation of a single complex [AuCl(H_m_L)]^m^ (m = 0–2), two complexes [AuCl(H_m_L)]^m^ and [AuCl(H_m-1_L)]^m−1^ (m = 1, 2), and three complexes (Equations (9)–(11)). The latter gives the best fit of calculated absorbance values to the experimental ones. It should be noted that the hydrazone substitutes three of four chloride anions in the coordination sphere; the last chloride likely remains or is substituted by hydroxyl anion at high pH values (this possibility is taken into account in the model’s Equations (4)–(7)). The calculated stability constants, as well as the stepwise protonation constants log K_a1_ and log K_a2_, of the complexes ([AuClL] + H^+^ ⇄ [AuCl(HL)]^+^ and [AuCl(HL)]^+^ + H^+^ ⇄ [AuCl(H_2_L)]^2+^, respectively; K_a1_ = [[AuCl(HL)]^+^] · [[AuClL]]^−1^ · [H^+^]^−1^, K_a2_ = [[AuCl(H_2_L)]^2+^] · [[AuCl(HL)]^+^]^−1^ · [H^+^]^−1^), and the constants of gold(III) binding by the protonated ligands, log K_f1_ and log K_f2_ ([AuCl_4_]^−^ + HL^−^ ⇄ [AuCl(HL)]^+^ + 3Cl^−^ and [AuCl_4_]^−^ + H_2_L ⇄ [AuCl(H_2_L)]^2+^ + 3Cl^−^, respectively; K_f1_ = [[AuCl(HL)]^+^] · [Cl^−^]^3^ · [[AuCl_4_]^−^] ^−1^ · [HL^−^] ^−1^; and K_f2_ = [[AuCl(H_2_L)]^2+^] · [Cl^−^]^3^ · [[AuCl_4_]^−^]^−1^ · [H_2_L]^−1^) are given in Table 1.
log K_a1_ = log β([AuCl(HL)]^+^)–log β([AuClL])(12)
log K_a2_ = log β([AuCl(H_2_L)]^2+^)–log β([AuCl(HL)]^+^)(13)
log K_f1_ = log β([AuCl(HL)]^+^)–log β(HL^−^)(14)
log K_f1_ = log β([AuCl(H_2_L)]^2+^)–log β(H_2_L)(15)

The complexes of gold(III) formed by pyridoxal-derived hydrazones are slightly more stable than those formed by **PLP**-derived hydrazones [39]. This small difference can probably be explained by the absence of an electron-withdrawing phosphate group, which favors the formation of the complex. However, the general similarity between the values of stability constants determined for the phosphorylated and dephosphorylated ligands indicates that the phosphate group does not participate in forming the complex with gold(III).

The UV-Vis spectra of the gold(III) complexes with **PL**-derived hydrazones that were calculated from the experimental data, together with the stability constants, are also similar to those reported for the complexes formed by **PLP**-derived hydrazones. They are also similar to the spectra of the protonated species (Figure 4a,b and Appendix A).

As in the previous paper [39], the relatively high values of the stability constant errors (Table 1) are because of the difficulties experienced while studying the complexation of hydrazones, as well as the similarity of the spectra of protonated and complex species. They are characteristic for such systems, as was noted by the authors of [50].

The speciation diagrams for all the studied complexes for the case of C(H[AuCl_4_]) = C(L) = 0.0001 mol L^−1^ are shown in Appendix A. In the physiological pH range (7.0 to 7.4), a mixture of [AuCl(HL)]^+^ and [AuClL] is the predominant species of gold(III). These results can be further used for evaluating the biological activity of gold(III) complexes with hydrazones derived from pyridoxal or calculating the equilibrium composition of the solution containing gold(III) complexes and biological macromolecules such as DNA and proteins.

The speciation diagrams also show that the hydrolysis of gold(III) species is hindered because of complexation with hydrazones.

In the next section, we will discuss the reasons why the UV-Vis spectra of complex gold(III) species look the way they look.

### 2.2. Geometry of Gold(III) Complexes

To interpret the UV-Vis spectra, DFT calculations of possible molecular models were performed. First, the initial geometry of the gold(III) complexes with **PL**-derived hydrazones was suggested to be analogous with the pyridoxal 5′-phosphate derivatives [40], where the phosphate group is replaced by hydroxyl. The subsequent optimization of the geometrical parameters and the calculation of the harmonic vibration frequencies of gold(III) complexes were carried out for the neutral [AuClL] molecule, monoprotonated [AuCl(HL)]^+^, and *bis*-protonated [AuCl(H_2_L)]^2+^ ions. An interesting observation can be made about the first protonation step: a proton can join two different groups, *viz.*, hydrazide nitrogen and heterocyclic nitrogen belonging to the pyridoxal moiety. Therefore, two possible configurations of monoprotonated species should be considered. The first one, [AuCl(HL)]^+^ 1, is a complex where a proton is added to the heterocyclic nitrogen of the pyridoxal residue, while the second one, [AuCl(HL)]^+^ 2, is a species where a proton is bound with hydrazide nitrogen. As an example, the considered molecular models of the complex with a **PL-F3H** hydrazone in different protonated states are given in Figure 5.

There are no imaginary frequencies for all considered molecular models, which means that all of them are minima on the potential energy surfaces (PESs). To test the chemical validity of the proposed gold(III) hydrazone complexes, the theoretical UV–Vis absorption spectra were calculated using TD DFT. The Cartesian coordinates of the *C*_1_ symmetry models of gold(III) complexes with **PL**-derived hydrazones, calculated IR intensities, and energies of vertical electronic transitions and oscillator strengths are presented in the Appendix A.

It is noteworthy that we neglected the possibility of phenolic oxygen protonation. A previous study [50] indicated that a completely ionized **PL-INH** hydrazone accepts protons in the following sequence: 1. Hydrazide nitrogen (log K = 10.25) 2. Heterocyclic nitrogen of pyridoxal residue (log K = 7.86) 3. Phenolic oxygen (log K = 4.41) 4. Heterocyclic nitrogen of isoniazid residue (log K = 2.95). In addition, forms of B_6_ vitamins tend to form zwitter-ionic species in aqueous solutions with proton transfer from a hydroxyl group to heterocyclic nitrogen [51,52,53,54,55], which likely applies to their hydrazone or Schiff base derivatives.

To simulate the shape of the experimental UV–Vis absorption spectra of the selected molecular models in the range of 200–500 nm, the individual bands were described by Lorentz curves with a half-width of 30 nm. As a result, in Figure 6, the TD DFT simulated UV–Vis absorption spectra are presented both for the complexes of **PL-F3H** and **PLP-F3H** [40] for comparison. For complexes that include other ligands, the observed and simulated UV–Vis absorption spectra are provided in the Figure 3 and Appendix A.

In the case of the observed UV–Vis spectra of all considered gold(III)–hydrazone complexes (see Figure 4a,b and Appendix A), the sequential protonation [AuClL]→[AuCl(HL)]^+^ 2→[AuCl(H_2_L)]^+2^ led to a noticeable absorption intensity change rather than a shifting of the spectral bands. For the complex with a **PL-F3H** hydrazone, the TD DFT spectra in the series of [AuClL]→[AuCl(HL)]^+^ 2→[AuCl(H_2_L)]^+2^ reproduced this tendency better than in the raw [AuClL]→[AuCl(HL)^+^ 1→[AuCl(H_2_L)]^+2^. Therefore, we can assume that the calculated UV-Vis spectrum of [AuCl(HL)]^+^ 2 agrees better with the experimental one than that of the [AuCl(HL)]^+^ 1 complex, which is protonated via heterocyclic nitrogen of **PL** residue.

A similar situation was observed for the gold(III) complexes with **PLP**-derived hydrazones: in the previous paper [40], the worst agreement was found between the calculated and experimental UV-Vis spectra of single-protonated complexes as well. However, they were more difficult cases, as the phosphate group can also accept protons. The most energetically preferable structure was chosen, which was a suboptimal decision, since a significant discrepancy between the calculated and experimental UV-Vis spectra was achieved as a result. Therefore, in the case of the gold(III) complexes with **PL**-derived hydrazones, we strongly recommend the [AuCl(HL)]^+^ 2 model (which considers a proton to be bound with a hydrazide group) for the interpretation of the chemical properties of a monoprotonated form in an aqueous solution, in spite of the [AuCl(HL)]^+^ 1 model being lower in energy.

The geometry of the gold(III) complexes with **PL**-derived hydrazones is reminiscent of that of the complexes formed by pyridoxal 5′-phosphate derivatives [40], which can be seen from Table 2 data. It presents the bond lengths between the gold(III) ion and donor atoms of the ligands.

The coordination sphere of gold(III) is close to a square planar geometry, as the AuNO_2_Cl fragment is close to the D_4h_ ideal reference polygon (see Figure 5). There is a minor difference between the geometries of the complexes formed by **PL-** and **PLP**-derived hydrazones, as follows from the DFT calculations. The comparison of the structural parameters of the AuO_2_NCl fragment of complexes with **PL-F3H** and **PLP-F3H** is presented in Table 3.

The phosphate group, which is remote to the donor atoms of the hydrazones, only slightly influences the complex’s structure. However, it is interesting to note that the incorporation of the phosphate group leads to a systematic increase in internuclear distances (**∆***r >* 0) in the AuO_2_NCl fragment (see Table 3), and this tendency is maintained for the remaining complexes as well. The elongation of distances in the coordination cavity of the complex for the phosphorylated ligand is the result of chemical bonds weakening between the gold cation and ligand due to the electron-withdrawing phosphate group effect. Thus, the presented molecular models do not contradict experimental results finding less stability of **PLP**-derived hydrazones, as mentioned above. The similarity in the structure of **PL-** and **PLP**-derived hydrazones, in general, also follows from the similarity of the simulated electronic spectra (see Figure 6).

The most noticeable changes in the coordination cavity geometry took place during the protonation of the complex. As in the previous paper [40], a significant contraction of Au–Cl (by ~0.04 Å) and elongation of Au–N (by > 0.01 Å) bonds is observed in the series of [AuClL]→[AuCl(HL)]^+^ 2→[AuCl(H_2_L)]^2+^. Adding a proton in the ligand leads to strong local electron density shift due to electrostatic attraction, which provides the changes in geometry in the coordination cavity and in the ligand.

### 2.3. The Nature of UV-Vis Absorption Spectra

The results of TD DFT calculations allow for an assignment of the experimental absorption bands. Table 4 summarizes the data on the selected vertical electronic transitions and oscillator strengths corresponding to the absorption band at about 300 and 350 nm for all discussed molecular models of gold(III) with hydrazones. The shapes of the selected molecular orbitals (MOs) with numbering are provided in Appendix A.

As in the case of gold(III) complexes with hydrazones derived from pyridoxal 5′-phosphate [40], the positions of the absorption band of the simulated spectra were close to the observed absorption maxima in the vicinity of 300 and 350 nm. In all cases, the absorption bands of the deprotonated species AuL^0^ correspond to π→π* transitions between ligand MO. The same applies to the complexes with phosphorylated ligands. The first protonation leads to two weak absorption bands arising (~370 and 350 nm) that could be assigned to a π→π* transition (S_0_→S_2_) and S_0_→S_4_ transition between the chlorine lone pair n_p_ and LUMO (denoted as σ*_p-d_) consisting of gold d-orbital and p-orbitals of N, Cl, and O atoms (see Appendix A). A significant contribution into the intensive band in the vicinity of 300 nm is provided by a σ_p_→ σ*_p-d_ transition, where σ_p_ is an MO formed by p-atomic orbitals of pyridoxal moiety. After the second protonation, the single band appears instead of two weak peaks in the 350 nm region, wherein the π→π* and σ_p_→ σ*_p-d_ transitions’ contributions are commensurate. The intensive band at 300 nm region mostly appeared due to a π→π* transition.

Figure 7 demonstrates frontier MOs of the complex with a **PL-F3H** hydrazone. The HOMO is of a π-character and localized in the ligand, while the LUMO is σ-antibonding and localized at the center of the coordination cavity, and, as a result, the HOMO→LUMO transition is inactive in gold(III) complexes’ absorption spectra. It should be noted that the composition of frontier MOs is the same for the remaining complexes.

## 3. Materials and Methods

### 3.1. Chemicals

The synthesis and spectral characteristics of **PL-F2H** and **PL-T2H** hydrazones are described in a previous paper [29], while the information about **PL-INH**, **PL-F3H**, and **PL-T3H** can be found in paper [47]. H[AuCl_4_]*3.4H_2_O (LenReaktiv, Russia, claimed Au content 49.11%) was used without additional purification. The concentration of perchloric acid used for titration purposes was determined alkalimetrically using the NaOH solution, which was standardized by Na_2_B_2_O_7_·10H_2_O. All the solutions for spectral studies were prepared using bidistilled water (κ = 3.6 μS cm^−1^, pH = 6.6).

### 3.2. UV-Vis Titration

UV-Vis spectral titration was performed using a Shimadzu UV1800 double-beam spectrophotometer (Shimadzu, USA, MA) within the spectral range of 200 to 500 nm and absorbance range of 0 to 2. The temperature was maintained at 298.2 ± 0.1 K using an external thermostat. Quartz cells with an optical path of 1.00 cm were used. All experiments were quadruplicated, at least. UV-Vis experiments consisted of the titration of 2.7 mL of the aqueous solution containing 202.4 µM H[AuCl_4_] and 299.4 µM HClO_4_ (total concentration of H^+^ = 501.7 µM) in water by an aqueous solution containing hydrazone (C(**PL-INH**) = 1.657 mM; C(**PL-F2H**) = 1.337 to 1.465 mM; C(PL-F3H) = 1.462 mM; C(PL-T2H) = 1.491 mM; C(PL-T3H) = 1.433 to 1.520 mM) and alkali (C(NaOH) = 9.93 to 10.84 mM). The volume of titrant added in one injection was 10 μL, and 25 to 30 titration points were acquired in total in each experiment.

Primary UV-Vis spectra were processed using the KEV software, version 0.7 [46] (Accessed (May 4, 2024) to calculate the equilibrium constants of the reaction between gold(III) species and hydrazones. The same software was used for drawing the speciation diagrams.

### 3.3. Quantum Chemical Calculation Details

The calculations of probable molecular models of gold(III) complexes with the hydrazones derived from pyridoxal were carried out for the singlet electronic state using the Gaussian program package, version 09W [56]. The equilibrium geometrical parameters and normal mode frequencies (Appendix A) were calculated using the hybrid DFT computational method B3LYP [57]. The energies of vertical electronic transitions and oscillator strengths (UV-Vis spectra) were calculated using the TDDFT [58] method CAM-B3LYP [59]. The two-component relativistic effective core potential (ECP60MDF) [60] was applied for the inner electronic shells of Au (1*s*^2^2*s*^2^2*p*^6^3*s*^2^3*p*^6^3*d*^10^4*s*^2^4*p*^6^4*d*^10^4*f*^14^). The valence shells (5*s*^2^5*p*^6^5*d*^10^6*s*^1^) were described by the (41s37p25d2f1g/5s5p4d2f1g) basis set cc-pVTZ-PP [61]. The H, C, N, O, P, and Cl atomic electronic shells were described by an all-electron cc-pVTZ basis set [62].To take into account the solvent effect of water, all calculations were performed using the polarizable continuum model (PCM) [63]. The visualization of ball-and-stick models and molecular orbitals was carried out using the ChemCraft program, version 1.8 [64].

## 4. Conclusions

The stability constants of complexes formed by tetrachloroaurate(III) and five pyridoxal-derived hydrazones in different protonated states were determined using spectrophotometry. Quantum chemical calculations combined with the experimentally determined stability constants, as well as the UV-Vis spectra of protonated species, allowed for the following observations:-The phosphate group does not participate in the complex formation; however, the electron-withdrawing effect may decrease the complex stability, where the derivatives of pyridoxal 5′-phosphate are involved;-During the first protonation stage, a proton may bind to different proton-accepting groups. In the case of pyridoxal hydrazones, these groups are the hydrazide nitrogen and heterocyclic nitrogen of pyridoxal residue. The derivatives of PLP have an additional protonation site, the phosphate group. The best agreement between the experimental and calculated UV-Vis spectra was achieved for both PL- and PLP-derived complexes if the first proton was considered to be bound with the nitrogen of the hydrazide group. The total energy value is misleading; based only on this value, the wrong choice of molecular model can be made;-The spectra of single- and double-protonated complexes of gold(III) are sophisticated. The transition between lone pairs of chlorine and LUMO consisting of both d-orbitals of gold(III) and p-orbitals of donor atoms contribute heavily to the observed peaks. On the contrary, the absorption bands in the spectra of deprotonated complexes are of π→π* character and involve mostly the ligand MO, with a negligible participation of metal orbitals.

It would be interesting to check in future studies how the change of donor atoms would affect the UV-Vis spectra of the complexes formed by gold(III).

## Figures and Tables

**Figure 1 ijms-25-05046-f001:**
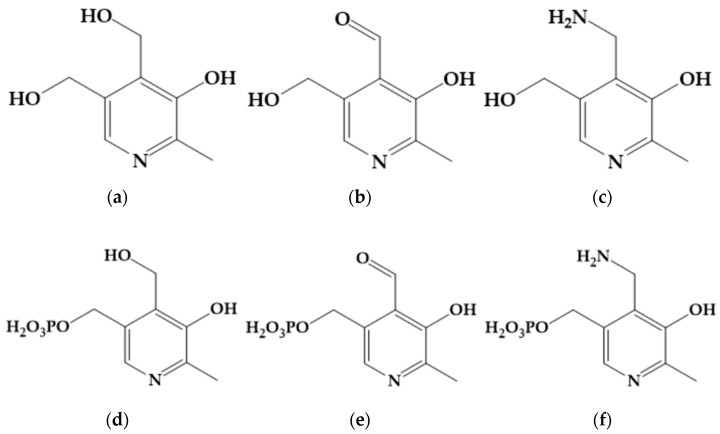
Pyridoxine (**a**), pyridoxal (**b**), pyridoxamine (**c**), pyridoxine 5′-phosphate (**d**), pyridoxal 5′-phosphate (**e**), and pyridoxamine 5′-phosphate (**f**).

**Figure 2 ijms-25-05046-f002:**
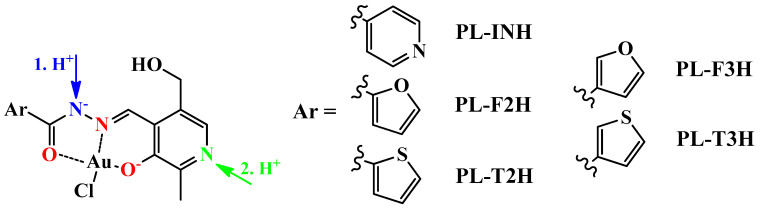
The assumed general structure of the neutral gold(III) complex with the following pyridoxal-derived hydrazones: **PL-INH** (Ar = 4-pyridinyl); **PL-F2H** (Ar = 2-furyl); **PL-F3H** (Ar = 3-furyl); **PL-T2H** (Ar = 2-thienyl); **PL-T3H** (Ar = 3-thienyl). Donor atoms involved in complex formation are marked with red, while acceptors of the first and second protons are marked with blue and green, respectively.

**Figure 3 ijms-25-05046-f003:**
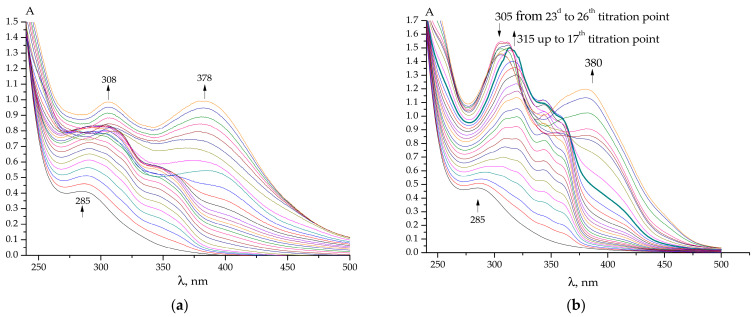
Examples of UV-Vis spectra acquired during the titration of (**a**) 202.4 µM H[AuCl_4_] + 299.4 µM HClO_4_ by 1.657 mM **PL-INH** + 10.34 mM NaOH; and (**b**) 200.9 µ H[AuCl_4_] + 298.9 µM HClO_4_ by 1.491 mM **PL-T2H** + 10.84 mM NaOH in water. Initial volume 2.7 mL, 25 to 30 titration points, titrant volume = 10 µL.

**Figure 4 ijms-25-05046-f004:**
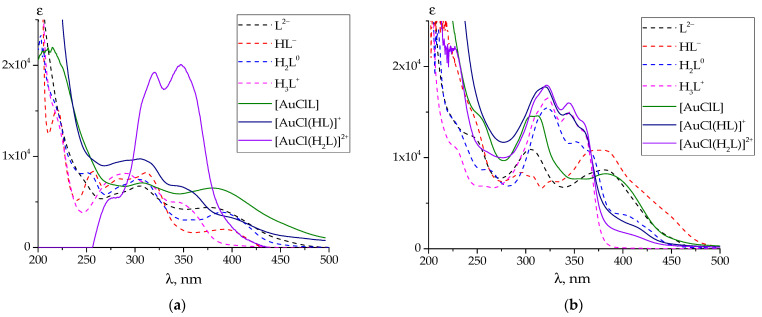
Calculated UV-Vis spectra of individual protonated and complex species of **PL-INH** (**a**) and **PL-T2H** (**b**). The protonated species spectra are adopted from [47].

**Figure 5 ijms-25-05046-f005:**
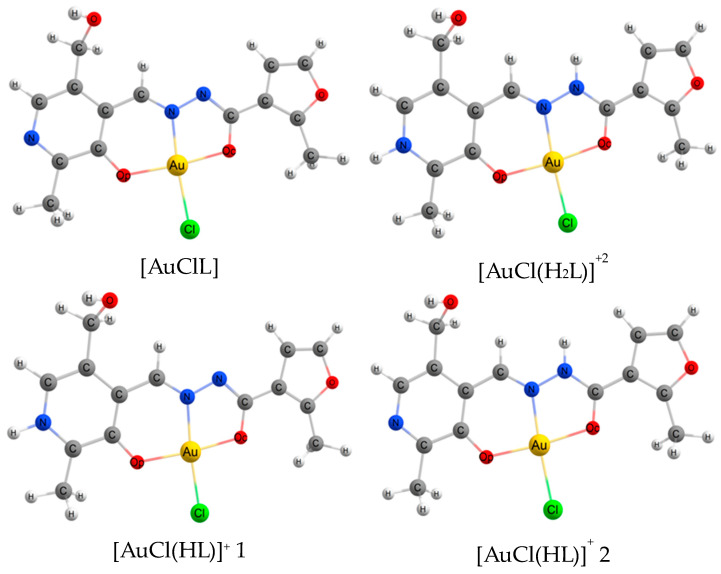
The *C*_1_ symmetry molecular models of neutral ([AuClL]), monoprotonated ([AuCl(HL)]^+^ 1, and [AuCl(HL)]^+2^) and *bis*-protonated ([AuCl(H_2_L)]^+2^) forms of complexes formed with **PL-F3H** hydrazone. O_p_ stands for oxygen of phenyl group in site 3 of pyridoxal moiety; O_c_ stands for carbonyl oxygen.

**Figure 6 ijms-25-05046-f006:**
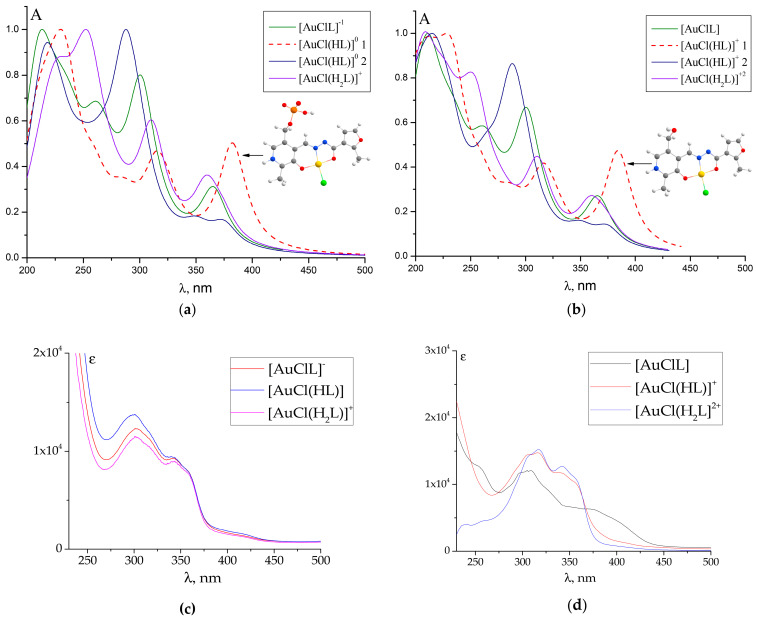
TD DFT (**a**,**b**) and experimental (**c**,**d**) UV-Vis spectra of different protonated forms of gold(III) complexes with **PLP-F3H** (**a**,**c**) and **PL-F3H** (**b**,**d**) hydrazones. The red dashed line spectra correspond to single-protonated complexes, where the proton is bound with heterocyclic nitrogen of a **PLP** ([AuCl(HL)] 1) and **PL** ([AuCl(HL)]^+^ 1) moiety. The theoretical individual bands were described by Lorentz curves with a half-width of 30 nm.

**Figure 7 ijms-25-05046-f007:**
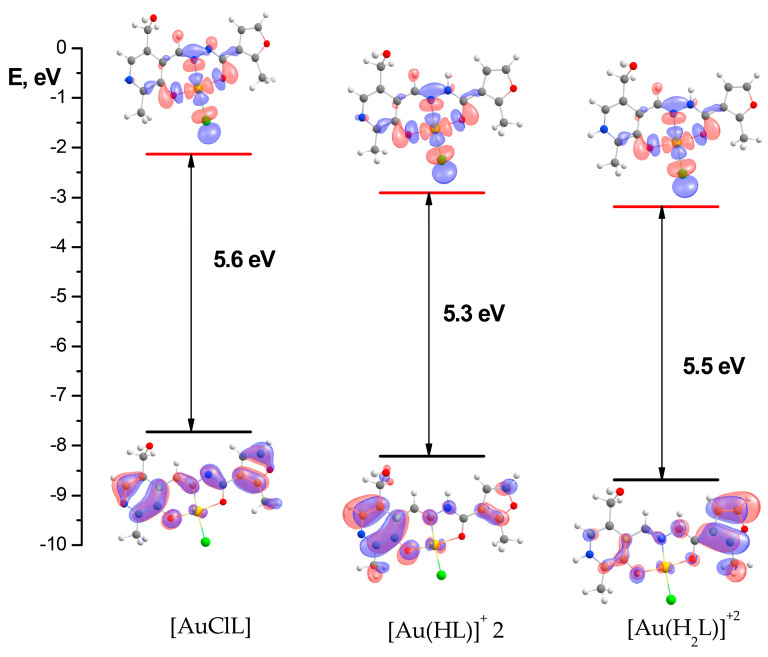
The HOMO–LUMO diagram for deprotonated [AuClL], monoprotonated [AuCl(HL)]^+^ 2, and *bis*-protonated [AuCl(H_2_L)]^+2^ forms of the gold(III) complex with a **PL-F3H** hydrazone.

**Table 1 ijms-25-05046-t001:** Stability constants of gold(III) complexes with hydrazones derived from pyridoxal in aqueous solution at *T* = 298.2 K, *I* ~0. The data for the complexes formed by PLP-derived hydrazones [39] are given for comparison.

Hydrazone	PL-INH	PL-F2H	PL-F3H	PL-T2H	PL-T3H
log β ([AuClL]) ^1^	13.3 ± 0.7	12.0 ± 1.2	12.9 ± 0.6	12.3 ± 0.9	12.5 ± 0.8
log β ([AuCl(HL)]^+^) ^1^	21.5 ± 0.8	19.3 ± 0.8	20.2 ± 0.4	20.0 ± 0.6	18.8 ± 0.3
log β ([AuCl(H_2_L)]^2+^) ^1^	25.9 ± 1.4	26 ± 2	25.6 ± 0.1	25.5 ± 1.1	24.3 ± 0.3
log K_a1_ ^2^	8.2 ± 1.1	7.3 ± 1.4	7.3 ± 0.7	7.7 ± 1.1	6.3 ± 0.9
log K_a2_ ^2^	4.4 ± 1.6	7 ± 2	5.4 ± 0.4	5.5 ± 1.3	5.5 ± 0.4
log K_f1_ ^3^	10.5 ± 0.7	7.9 ± 0.8	8.2 ± 0.6	8.2 ± 0.6	6.8 ± 1.0
logK_f2_ ^3^	6.6 ± 1.4	8 ± 2	8.2 ± 0.3	7.9 ± 1.3	6.0 ± 0.8
**Hydrazone** [39]	**PLP-INH**	**PLP-F2H**	**PLP-F3H**	**PLP-T2H**	**PLP-T3H**
log β ([AuClL]^−^)	11.2 ± 0.5	12.4 ± 0.9	12.0 ± 0.5	13.1 ± 0.8	12.5 ± 1.0
log β ([AuCl(HL)])	17.8 ± 0.8	18.5 ± 0.6	18.6 ± 0.6	20.3 ± 0.6	18.4 ± 0.3
log β ([AuCl(H_2_L)]^+^)	23.7 ± 0.9	24.1 ± 0.7	24.7 ± 0.7	26.0 ± 0.5	24.2 ± 0.2
log K_a1_	6.6 ± 0.9	6.1 ± 1.1	6.6 ± 0.8	7.2 ± 1.0	5.9 ± 1.0
log K_a2_	5.9 ± 1.2	5.6 ± 0.9	6.1 ± 0.9	5.7 ± 0.8	5.8 ± 0.4
log K_f1_	6.4 ± 0.9	7.1 ± 0.6	7.2 ± 0.6	8.8 ± 0.6	6.9 ± 1.0
logK_f2_	4.1 ± 0.9	4.4 ± 0.7	5.0 ± 0.8	6.2 ± 0.7	4.9 ± 0.4

^1^ The errors are the half-widths of the confidence interval at a confidence probability of 0.95 and sample size of 4 to 6 experiments; ^2^ The errors are the square roots taken from the sum of squared errors of log β([AuCl(H_m_L)]^m^) that are used for calculations; ^3^ The errors are the square roots taken from the sum of squared errors of log β([AuCl(H_m_L)]^m^) and protonation constants of hydrazones that are used for calculations.

**Table 2 ijms-25-05046-t002:** Equilibrium distances (Å) of AuO_2_NCl fragment in gold(III) hydrazone complexes, obtained by DFT/B3LYP calculations.

Distance	PL-F3H	PL-F2H	PL-T3H	PL-T2H	PL-INH
*r*(Au–Cl)	2.333 *	2.332	2.332	2.332	2.330
	2.302	2.301	2.302	2.302	2.300
	2.293	2.292	2.293	2.292	2.291
*r*(Au–N)	1.987	1.989	1.988	1.989	1.987
	1.994	1.996	1.994	1.995	1.995
	1.999	2.001	1.999	2.000	2.000
*r*(Au–O_p_) **	1.980	1.978	1.980	1.978	1.979
	1.969	1.968	1.969	1.968	1.967
	1.971	1.969	1.971	1.969	1.968
*r*(Au–O_c_) **	2.002	2.005	2.002	2.004	2.004
	2.030	2.032	2.030	2.030	2.038
	2.013	2.015	2.014	2.013	2.021
*r*(O_p_…O_c_)	3.979	3.980	3.979	3.978	3.979
	3.996	3.996	3.995	3.994	4.001
	3.980	3.981	3.980	3.979	3.985

* The three horizontal rows for each distance correspond to [AuClL], [AuCl(HL)]^+^ 2, and [AuCl(H_2_L)]^2+^ forms, respectively. [AuCl(HL)]^+^ 2 denotes the monoprotonated complexes, where the proton is bound with hydrazide nitrogen. ** O_p_ stands for oxygen of a phenyl group in site 3 of pyridoxal moiety; O_c_ stands for carbonyl oxygen.

**Table 3 ijms-25-05046-t003:** Comparison of equilibrium distances (Å) of AuO_2_NCl fragment of gold(III) complexes formed by **PL-F3H** and **PLP-F3H** hydrazones, obtained by DFT/B3LYP calculations.

Distance	PL-F3H	PLP-F3H [40]	∆*r ***,* Å
*r*(Au–Cl)	2.333 *	2.334 *	0.001
	2.302	2.304	0.002
	2.293	2.294	0.001
*r*(Au–N)	1.987	1.987	0.000
	1.994	1.995	0.001
	1.999	2.000	0.001
*r*(Au–O_p_) **	1.980	1.981	0.001
	1.969	1.970	0.001
	1.971	1.971	0.000
*r*(Au–O_c_) **	2.002	2.003	0.001
	2.030	2.032	0.002
	2.013	2.015	0.002
*r*(O_p_…O_c_)	3.979	3.980	0.001
	3.996	3.998	0.002
	3.980	3.983	0.003

* The three horizontal rows for each distance correspond to diprotonated, monoprotonated, and *bis*-protonated forms, respectively. The monoprotonated form corresponds to a complex, where the proton is bound with hydrazide nitrogen. ** O_p_ stands for oxygen of a phenyl group in site 3 of pyridoxal moiety; O_c_ stands for carbonyl oxygen. *** **∆***r = r(PL-F3H)*–*r(PL-F3H).*

**Table 4 ijms-25-05046-t004:** Selected vertical electronic transitions (UV–Vis absorption spectra), calculated using the TD DFT/CAM-B3LYP method for gold(III) complexes, with hydrazone-derived pyridoxal in aqueous solution.

Hydrazone	Protonated Form	Excited State	λ_cal_ (nm)	Oscillator Strength(*f*)	Composition *	Character
**PL-F3H**	[AuClL]	S_3_	365.97	0.2165	93 ** → 95 (79%), 92 → 95 (11%)	π→π*
		S_6_	301.06	0.5301	92 → 95 (56%), 91 → 95 (30%)	π→π*
	[AuCl(HL)]^+^ 2	S_2_	374.6	0.0819	93 → 95 (79%), 88 → 94 (12%)	π→π*
		S_4_	349.85	0.0642	88 → 94 (85%), 93 → 95 (15%)	*n_p_*→ σ**_p-d_*
		S_9_	287.64	0.6095	92 → 95 (31%), 91 → 94 (27%)	π→π*
	[AuCl(H_2_L)]^2+^	S_4_	359.37	0.1719	90 → 94 (43%), 92 → 95 (25%)	*n_p_*→ σ**_p-d_*
		S_6_	311.17	0.342	93 → 95 (46%), 92 → 95 (37%)	π→π*
**PL-F2H**	[AuClL]	S_3_	371.51	0.2062	89 → 91 (61%), 88 → 90 (24%)	π→π*
		S_6_	308.44	0.5742	88 → 91 (81%)	π→π*
	[AuCl(HL)]^+^ 2	S_2_	377.82	0.117	89 → 91 (77%), 84 → 90 (11%)	π→π*
		S_4_	352.67	0.0803	84 → 90 (87%), 89 → 91 (13%)	*n_p_*→ σ**_p-d_*
		S_9_	290.76	0.7339	87 → 90 (39%), 88 → 91 (17%)	σ *_p_*→ σ**_p-d_ */π→π*
	[AuCl(H_2_L)]^2+^	S_4_	362.1	0.2309	86 → 90 (42%), 89 → 91 (37%)	*n_p_*→ σ**_p-d_* /π→π*
		S_6_	305.19	0.7245	88 → 91 (67%), 89 → 92 (14%)	π→π*
**PL-T3H**	[AuClL]	S_3_	366.64	0.2199	93 → 95 (92%)	π→π*
		S_6_	300.26	0.5547	92 → 95 (83%)	π→π*
	[AuCl(HL)]^+^ 2	S_2_	377.02	0.0906	93 → 95 (79%), 87 → 94 (11%)	π→π*
		S_4_	351.09	0.0639	87 → 94 (86%), 93 → 95 (14%)	*n_p_*→ σ**_p-d_*
		S_9_	284.03	0.7723	92 → 95 (31%), 91 → 94 (27%)	π→π*
	[AuCl(H_2_L)]^2+^	S_4_	359.82	0.1824	89 → 94 (42%), 93 → 95 (37%)	*n_p_*→ σ**_p-d_* /π→π*
		S_7_	297.69	0.4448	92 → 95 (36%), 91 → 95 (25%)	π→π*
**PL-T2H**	[AuClL]	S_2_	371.3	0.2387	93 → 95 (74%), 92 → 94 (15%)	π→π*
		S_6_	308.01	0.5862	92 → 95 (87%)	π→π*
	[AuCl(HL)]^+^ 2	S_2_	377.64	0.1172	93 → 95 (76%), 87 → 94 (10%)	π→π*
		S_4_	351.66	0.0801	87 → 94 (88%), 93 → 95 (12%)	*n_p_*→ σ**_p-d_*
		S_9_	292.24	0.5032	91 → 94 (40%), 84 → 94 (13%)	σ *_p_*→ σ**_p-d_*
	[AuCl(H_2_L)]^2+^	S_4_	361.32	0.2194	89 → 94 (48%), 93 → 95 (36%)	*n_p_*→ σ**_p-d_* /π→π*
		S_6_	307.83	0.6715	92 → 95 (53%), 93 → 96 (19%)	π→π*
**PL-INH**	[AuClL]	S_2_	368.06	0.1511	92 → 94 (79%), 91 → 93 (10%)	π→π*
		S_6_	299.08	0.3955	91 → 94 (85%)	π→π*
	[AuCl(HL)]^+^ 2	S_2_	381.75	0.0681	92 → 94 (72%), 86 → 93 (13%)	π→π*
		S_4_	357.25	0.0524	86 → 93 (81%), 92 → 94 (16%)	*n_p_*→ σ**_p-d_*
		S_11_	272.04	0.4007	91 → 93 (31%), 90 → 94 (17%)	σ *_p_*→ σ**_p-d_* /π→π*
	[AuCl(H_2_L)]^2+^	S_4_	361.84	0.1542	92 → 94 (74%), 88 → 93 (16%)	π→π*
		S_12_	260.17	0.3955	89 → 93 (22%), 86 → 94 (18%)	π→π*

* The composition includes the first two most significant (%) transitions. The transitions with contributions below 10% are omitted. ** A visualization of molecular orbitals according to the numbers is presented in Appendix A.

## Data Availability

Data are contained within the article or Appendix A.

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
