# Peer review of "Complexes of Gold(III) with Hydrazones Derived from Pyridoxal: Stability, Structure, and Nature of UV-Vis Spectra"

_ijms, 2024, doi:10.3390/ijms25095046_

Round 1
Reviewer 1 Report (New Reviewer)
Comments and Suggestions for Authors
The authors present theoretical and experimental investigation of complexes of gold(III) with hydrazones derived from pyri-2 doxal. Authors aim to answer the question whether the hosphine group participates in complex formation between the hydrazone and gold. Unfortunately, the novelty and aim of tis research is not explained enough. Taking this into account, the aim of the work is not clear. Authors should improve the justification of this chosen topic more evidently in the introduction part.
Abstract should be modified to describe the results of the research more, and avoid general introduction to the topic.
The description of hydrazones should be expanded by addition of not only its’ biological properties, but by highlighting the main chemical bond, that induces the bioreactivity. Furthermore, authors could include some hydrazone formation descriptions (https://doi.org/10.1016/j.synthmet.2014.03.009; https://doi.org/10.1016/j.synthmet.2014.08.009; https://doi.org/10.1080/15685551.2013.840499).
I would recommend the publication of this manuscript after minor revision.
Comments on the Quality of English Language
The language is good, moderate editing is necessary.
Author Response
Dear Sir,
we appreciate your positive feedback, your time, and your efforts devoted to helping us improve our contribution. Please, find the answers to your specific comments below:
- Q.: "Authors should improve the justification of this chosen topic more evidently in the introduction part."
A.: The Introduction section is revised to justify the chosen topic and methods of choice more clearly. - Q.: "Abstract should be modified to describe the results of the research more, and avoid general introduction to the topic."
A.: Preparing the manuscript, we have followed the guidelines provided by IJMS. An abstract structure that includes more or less detailed background is recommended by the Journal. - Q.: "The description of hydrazones should be expanded by addition of not only its’ biological properties, but by highlighting the main chemical bond, that induces the bioreactivity. Furthermore, authors could include some hydrazone formation descriptions (https://doi.org/10.1016/j.synthmet.2014.03.009; https://doi.org/10.1016/j.synthmet.2014.08.009; https://doi.org/10.1080/15685551.2013.840499)."
A.: Thanks a lot for the recommendations. We have read the great papers by Prof. Gražulevičius with strong interest. They are even more of interest to us as we currently study some electrochemical and fluorescent properties of our hydrazones in the solid phase. Without a doubt, we will be able to find many useful ideas in the articles of this group.
The corrections made according to your comments are highlighted with green.
Reviewer 2 Report (New Reviewer)
Comments and Suggestions for Authors
In this manuscript, authors examine the involvement of the phosphate group in the formation of a complex between hydrazones that are derivatives of pyridoxal-phosphates and gold(III). It is clear from the results that phosphate does not bind to gold, although the indirect influence of the phosphate on the complexation does exist.
In future work on a similar topic, I suggest that the authors test other DFT functionals and compare the results they obtained using B3LYP.
I believe that this manuscript could be published as is, although some of the aspects of the presentation of the results could be more elegant. However, that is more a matter of taste.
Author Response
Dear Reviewer,
we appreciate your time spent, your efforts, and your valuable suggestion to use different methods and basis sets to create a necessary benchmark in our future papers. We will do this in our further work.
This manuscript is a resubmission of an earlier submission. The following is a list of the peer review reports and author responses from that submission.
Round 1
Reviewer 1 Report
Comments and Suggestions for Authors
In this manuscript gold(III) complexes with hydrazones derived from pyridoxal have been investigated. The structure of the complexes, the stability constants, the Uv-vis spectra have been studied experimentally and computationally. I believe that the manuscript could be accepted for publication if the following issues are solved.
-the manuscript is in my opinion vey difficult to read, considering the similarities between the abbreviations used for the different “ligands”. Furthermore, the structure of the formed complexes is reported for the first time only in the DFT section (figure 4), so it is not clear from the beginning how the ligands coordinate to the gold centers. I suggest to add a general sketch for the complexes in their neutral, mono- and di-protonated form.
-equations 1-11 are chemical reactions/equilibria so that they must be written correctly (with arrows, correct charges for the species, and so on).
-L identifies the ligand but the ligand has been deprotonated so, that it is L(2-). HL is the protonated form: where is H located? The authors in the DFT calculations conclude that the H hydrogens are located on the nitrogen atoms. How can they exclude that the phenolate substituent is not reactive as well? Are they sure that it remains coordinated to the metal?
-HL or H2L should be written in parenthesis when coordinated to gold since from a coordination point of view the AuHL notation is not correct and the H hydrogen can be confused with an hydrido ligand. For this reason change every formulae in the text with Au(HL) or Au(H2L).
-The square planar coordination of gold(III) is completed by a chloride ligand and in my opinion it would be better to include this ligand in the formula of the complexes AuClL, AuCl(HL)+, and so on.
-from a coordination point of view, it is not correct to indicate the complex as AuCl(PL-F3H) (and so for the other ones). PL-F3H is not the ligand but the proligand, it must be deprotonated to be coordinated to the metal.
Author Response
Please, see the attached file.

Reviewer 2 Report
Comments and Suggestions for Authors
This article could be improved and I will justify my decision below.
While from the technical point of view the study would be acceptable, the level of scientific novelty is not appropriate for Q1 journal with high IF such as IJMS. The Authors haven’t obtained any new compounds (all of the synthesis were previously described in [20] and [31]) nor provided any substantial new results apart from routine UV-Vis measurements and DFT/B3LYP calculations of few simple organometallic compounds. This study is just minor extension of the previous works of the same authors. Compounds or Spectroscopy Journal, both of MDPI, should be a better choice for this research note.
Detailed comments can be found below.
In the whole abstract there is not a single word about the gold(III), while it is the major component of the studied systems. Why?
Line 24, a figure should be created presenting those structures.
Line 49, lgβ should be defined
Line 83, “in general” should be removed
Figures 4 and 5, why the theoretical spectra are not compared with the experimental ones?
The Authors have obtained, from UV-Vis, the stability constants of the complexes. The Authors have also calculated the structures of the modeled systems, including the free energies. Why the Authors haven’t compared the theoretical and experimental values of K? I am aware that it would require to calculate the free energies of the Au(III) and the ligands, including the organic molecules, but this should be done.
Line 361, why is data sharing not applicable?
Besides, the article is not in the scope of the Special Issue that it was submitted to.
Author Response
Please, see the attached file.

Reviewer 3 Report
Comments and Suggestions for Authors
The presented paper continues the study on Au3+ complexes with hydrazones derived from pyridoxal 5′-Phosphate that was published in IJMS in the last year. It reports on the stability of Au3+ complexes derived from pyridoxal and isoniazid, 2-furoylhydrazide, thiophene-2-carbohydrazide, 2-methylfuroyl-3-hydrazide, and thiophene-3-carbohydrazide. The comparison was made with the previous data, and the role of the phosphate group in the complex formation with hydrazones was deduced. It appeared that phosphate group does not participate in the complex formation, because its presence only slightly influences the complex structure. Its presence results in an insignificant increase of the internuclear distances in AuO2NCl fragment due to the weakening of chemical bonds between gold cation and ligand. The UV-Vis experiments were carried out and the calculations of energies of vertical electronic transitions and oscillator strengths were calculated by the TDDFT method.
In general, this work represents an interesting study involving both experiment and theory. The obtained results can be used in drug discovery. The manuscript is well-structured and carefully written. No grammatical issues detected. In my opinion, this work is worth publishing in IJMS. In a whole, there is not very much to improve, the work seems fine as it is, though, some minor recommendations can be proposed, please, vide infra.
1. I suggest to add the figure representing the hydrazones derived from pyridoxal 5′-phosphate (from previous paper in IJMS) and the corresponding gold(III) complexes with these hydrazones for the sake of comparison. Table I should also be extended to PLP-X.
21. Ref. 29. Page is missing.
32. Define all notifications (log β, log K, …)
43. Fig. 4. Some oxygen symbols are incorrect (these are Op, Oc, or something like this).
54. Geometry optimizations were performed with the DFT/B3LYP. Were the dispersion corrections included? If not, why not? These are important for complexes.
65. NMR spectra?

Author Response
Please, see the attached file

Round 2
Reviewer 1 Report
Comments and Suggestions for Authors
The manuscript has been improved according to the suggestions of the referee. There is still some space for improvement in my opinion.
First of all, it will be better to show or indicate the Ar groups directly in Figure 2 and not only in the caption.
Speciation studies are important but it would be interesting to isolate one of the proposed complexes of formula [AuCl(L)]. The isolation of the complex, together with protonation studies in solution, followed for example with NMR techniques would help in supporting the findings of the manuscript. I understand that this is outside the scope of the present manuscript but it would support the conclusions and strengthen the paper.
Author Response
According to your recommendation, we added the Ar groups directly to Figure 2. We are grateful for your positive feedback and appreciate the suggestion for further work on this project.
Reviewer 2 Report
Comments and Suggestions for Authors
The Authors haven’t convinced me about the quality of their work and significance of the results obtained. Therefore, sadly, I have to suggest the rejection despite the changes made in the manuscript. My major comments are the same as in the previous round, but the Authors have already denied to include them (calculations of K from the DFT thermochemistry). I find this manuscript technically OK, but in my opinion it is more suitable for the other journals of MDPI. Compounds or Spectroscopy Journal, both of MDPI, should be a better choice for this research note.
Author Response
We respectfully appreciate your feedback. However, we respectfully disagree with your opinion. The quantum chemical calculations of the stability constants are still in the early stages and may not achieve chemical accuracy. These calculations are typically utilized when the experimental determination of the complex stability is unavailable or too complicated. In our study, we determined the stability constants using experimental techniques such as spectrophotometry. We find it challenging to justify the investment of time and resources into computations that may not yield stability constant values consistent with our experimental findings. The potential margin of error in these calculations is significant, with the lowest discrepancy likely being 2 log units and the upper limit of error being indeterminate. While this is an intriguing area for future exploration, we may consider focusing on it in our upcoming research endeavors.